# Bimodal Microstructure Obtained by Rapid Solidification to Improve the Mechanical and Corrosion Properties of Aluminum Alloys at Elevated Temperature

**Irena Paulin [1,*], Črtomir Donik [1], Peter Cvahte [2] and Matjaž Godec [1]**

[1]  Institute of Metals and Technology, Lepi pot 11, 1000 Ljubljana, Slovenia; crtomir.donik@imt.si (Č.D.); matjaz.godec@imt.si (M.G.)
[2]  IMPOL 2000 d.d., Partizanska ulica 38, 2310 Slovenska Bistrica, Slovenia; peter.cvahte@impol.si
[*]  Correspondence: irena.paulin@imt.si

**Abstract:** The demand for aluminum alloys is increasing, as are the demands for higher strength, with the aim of using lighter products for a greener environment. To achieve high-strength, corrosion-resistant aluminum alloys, the melt is rapidly solidified using the melt-spinning technique to form ribbons, which are then plastically consolidated by extrusion at elevated temperature. Different chemical compositions, based on adding the transition-metal elements Mn and Fe, were employed to remain within the limits of the standard chemical composition of the AA5083 alloy. The samples were systematically studied using light microscopy, scanning electron, and transmission microscopy with electron diffraction spectrometry for the micro-chemical analyses. Tensile tests and Vickers microhardness were applied for mechanical analyses, and corrosion tests were performed in a comparison with the standard alloy. The tensile strength was improved by 65%, the yield strength by 45% and elongation by 14%. The mechanism by which we achieved the better mechanical and corrosion properties is explained.

**Keywords:** aluminum alloy AA5083; rapid solidification; melt spinning; high-strength aluminum; extrusion; bimodal microstructure; precipitations

## 1. Introduction

Aluminum alloys are increasingly popular in the transport industry because of their good corrosion resistance, very good strength-to-weight ratio, and the consequent reduced $CO_2$ footprint. The most often used alloys with the best mechanical properties are the 2xxx and 7xxx series, which possess high strength and ductility, which benefits from the precipitation strengthening, but have a relatively poor corrosion resistance. In contrast, alloys from the 5xxx series have a good corrosion resistance, but only moderate strength.

High-strength aluminum alloys that are stable at elevated temperatures (above 200 °C) cannot be produced commercially using conventional routes, if their high strength properties are to be based only on precipitation strengthening. It is common that a distinctive loss of mechanical properties is observed at moderate temperatures (above 150 °C) due to over-aging effects. Aluminum alloys that are resistant to high temperature must have a stable microstructure, which is achieved by having stable, incoherent particles in the matrix. These represent barriers to the sliding of dislocations and can be introduced by a rapid solidification of the melt, where smaller insoluble particles remain in the matrix. These particles prevent the movement of the sliding planes and the moving/migration of dislocations. To improve the thermal stability of aluminum alloys, low-diffused transition metals (TMs) such as Fe, Mn, Cr, Ni, V, Co, and/or Mo must be introduced.

Previous studies confirmed the benefits of TM elements for high-strength aluminum alloys at elevated temperature through the addition of specific TM elements, either individually [1] or in pairs [2]. Some studies were performed with fixed amounts of one

or two TM elements and the variation of a third element [3]. All the studies used pure aluminum with a specific addition of TM elements. Very few authors [2,4] highlight the importance of using scrap aluminum with contamination in the form of elements mixtures, which have a role in improving thermal stability [5–7]. The diffusivity and solubility of such elements in aluminum is very low, and during conventional production routes, hard and brittle particles of Al-TM intermetallic phases are formed. These large phases reduce the mechanical properties [8,9]. The solution is to dissolve the elements in an aluminum matrix or disperse them in fine intermetallic particles, which can be produced by increasing the solidification rate through melt spinning (cooling rate $10^4$–$10^6$ K/s) or atomization (cooling rate $10^2$–$10^4$ K/s) [10]. Rapid solidification (RS) leads to interesting features, like a significant increase of the alloying elements' solid solubility, a greater refinement of the microstructural grains and the formation of a variety of non-equilibrium aperiodic phases. [11] The equilibrium solubility of TM elements in an aluminum lattice is very low, commonly around 0.03 at. %, and consequently they cannot bring about any effective strengthening with a conventional thermal treatment. The microstructure of rapidly solidified alloys consists of supersaturated solid solution of alloying elements in aluminum, with stable, metastable and quasicrystalline intermetallic phases. [3,12]

The previous investigations and trials attempting to improve the mechanical properties at elevated temperatures for aluminum alloys tended to use pure alloy elements and studied and improved only some of the properties, mostly mechanical, but did not produce materials that would improve a range properties, including corrosion, at the same time. Although it is known that, normally, a major improvement in the mechanical properties leads to moderate corrosion properties [13], and vice versa. Excessive amounts of TM elements have also been used to realize a large increase in the mechanical properties. But the explanation of the strengthening mechanism did not make it clear that, for example, with an excessive amount of added TM elements, despite the high cooling rate, larger Al-TM phases can occur, which negatively affect the corrosion rate.

The main objective of our study was to importantly improve the mechanical and corrosion properties of the aluminum alloy AA5083, which is known as a good corrosion-resisting material with moderate strength. With minor alloy modifications and a rapid solidification route we intended to modify the microstructure and consequently improve the mechanical and corrosion properties at elevated temperature which would classified these materials as high-strength aluminum alloys.

## 2. Materials and Methods

The commercial aluminum alloy AA5083 (EN AW 5083) was modified with the alloying elements, rapidly solidified, and then isostatically pressed at room temperature, with a final hot extrusion (Figure 1).

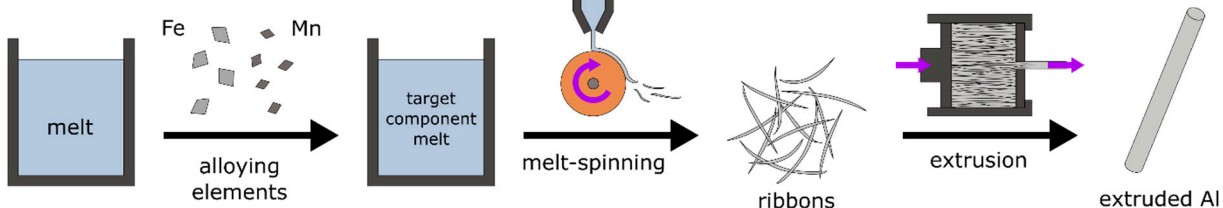

**Figure 1.** Scheme of the material's preparation.

The chemical compositions of the standard AA5083 aluminum alloy and the modified samples with the addition of only Mn as well as Fe and Mn are presented in Table 1. The most important feature is to apply the modification of the elements up to the allowed tolerance limit according to standard. This is important for industry, because, for example, the transport industry is extremely strict regarding safety standards and is not permitted to introduce new materials until there has been at least 10 years of testing. By staying within

the limits of the standard chemical composition, the new, improved material can be more readily accepted.

**Table 1.** Chemical composition of studied materials (in wt. %).

| Samples | Si | Mn | Fe | Zn | Mg | Ti | Al |
|---|---|---|---|---|---|---|---|
| Standard AA5083 [14] | max. 0.40 | 0.40–1.00 | max. 0.40 | max. 0.25 | 4.00–4.90 | max. 0.15 | balance |
| Master AA5083 alloy | 0.20 | 0.52 | 0.34 | 0.024 | 4.3 | 0.016 | balance |
| RS AA5083 | 0.20 | 0.52 | 0.32 | 0.024 | 4.3 | 0.016 | balance |
| RS AA5083 with Mn | 0.20 | 1.00 | 0.31 | 0.024 | 4.3 | 0.016 | balance |
| RS AA5083 with Fe and Mn | 0.20 | 1.00 | 0.45 | 0.024 | 4.3 | 0.016 | balance |

Melt spinning was used to prepare rapidly solidified ribbons. The aluminum melt was superheated to 1400 °C and cast through small orifice onto a water-cooled copper wheel rotating with a circumferential speed of 50 m/s.

The obtained ribbons were cold compacted under pressure of 200 MPa into billets of 24 mm diameter and approximately 70 mm height. The billets were preheated at the temperature of 420 °C for 15 min prior to the extrusion. The reduction was from 24 mm to 6 mm and the reduction ratio was 16. For comparison, an industrially-extruded standard AA5083 alloy (our master alloy) was used.

The microstructure of the obtained ribbons was first characterized by light microscopy (LM). The samples were prepared by standard metallographic procedure by grinding and polishing with 1-μm surface finishes. To reveal the microstructure for LM the samples were etched using the Weck two-stage etching process. The LM investigation was performed with a Microphot FXA (Nikon Instruments Inc., Tokyo, Japan), an Olympus DP73 camera (Olympus Europe Holding GMBH, Hamburg, Germany) and the Stream Motion (Olympus, Tokyo, Japan) computer program. The microstructure investigations of the ribbons and the extruded materials were performed by scanning electron microscopy (SEM, JEOL FESEM JSM 6500F, (JEOL, Tokyo, Japan)) using electron back-scatter diffraction (EBSD, camera HKL Nordlys II with Channel 5 software (Oxford Instruments HKL, Hobro, Denmark)) for the grain size and the orientation, and energy-dispersive spectroscopy (EDS, INCA ENERGY 400, Oxford Instruments, High Wycombe, UK) for the micro-elemental analysis. Samples of the ribbons were prepared with an ion slicer that was perpendicular to the ribbon. The extruded samples were prepared perpendicular to the extrusion direction by grinding and mechanically polishing with 2 min oxide polishing suspension (OPS). The microstructures of selected ribbons were characterized by high-resolution transmission electron microscopy (TEM, JEOL JEM-2100, JEOL, Tokyo, Japan) in bright field (BF) mode coupled with an EDS analyzer (INCA, Oxford Instruments, High Wycombe, UK).

The Vickers microhardness was measured with a TUKON 2100B (Instron, Norwood, MA, US) instrument. For testing of the ribbons, a 0.025 kgf load was used, and for the extruded material a 0.5 kgf load was used. The microhardness measurements were performed on every sample 5 times and the average values are presented.

The chemical analyses were performed using optical emission spectrometry with an inductive coupled plasma (ICP-OES Agilent 720, (Agilent, Santa Clara, CA, USA)).

The corrosion testing of the extruded samples was performed using electrochemical impedance spectroscopy. An exposed 1 cm$^2$ of extruded Al sample was prepared for the electrochemical testing in 3.5% NaCl solutions. All the samples were mechanically prepared with SiC paper up to 2400 grit and polished to a mirror finish. Before the electrochemical measurements, the samples were washed and rinsed with acetone and washed in deionized water and dried in air. All the solutions for the experiment were made using MERCK (Merck KGaA, Darmstadt, Germany) chemicals and deionized water. A three-electrode system was used for the measurements, an aluminum electrode (working electrode), a saturated calomel electrode (SCE) as the reference electrode and a Pt mesh

as the counter electrode. All the electrochemical measurements were performed using a Potentiostat/Galvanostat BioLogic SP 300 (Biologic, Seyssinet-Pariset, France) with EC-Lab V11.27 software (Biologic Science Instruments, Seyssinet-Pariset, France). The measurements of all the potentiodynamic polarizations were conducted at a scan rate of 1 mV/s. The data were gathered for the determination of the electrochemical parameters: Corrosion current density $i_{corr}$ and corrosion potential $E_{corr}$ to make the comparison of the extruded samples.

Mechanical tests were performed on an INSTRON 1255 machine (500 kN, Instron, High Wycombe, Buckinghamshire, UK) at room temperature with a constant velocity corresponding to an initial strain rate $10^{-2}$ s$^{-1}$. For each group of samples, at least five samples were tested. Based on the results the tensile strength ($R_m$), yield strength ($R_{p0.2}$) and elongation (A) were recorded. The results are presented as average values.

## 3. Results and Discussions

To achieve the high strength and corrosion resistance of the aluminum alloy, the melt is rapidly solidified in the form of ribbons and further plastically consolidated using extrusion at elevated temperature. Different chemical compositions of the alloys were prepared and melt spun. The products of melt spinning are typically ribbons, approximately 3 mm wide and 50–120 μm thick, as shown in Figure 2a. The melt-spun ribbons were characterized in the longitudinal direction. An etched LM image reveals the microstructure of the RS aluminum alloy (Figure 2b). Columnar grains appeared on the side where the melt touched the copper wheel. The ribbon's surface on this side is flat and the precipitates are very small and not directly visible in LM micrographs. They are seen as etched pits due to the corrosion attack of the small precipitates. The ribbon's surface on the air side is not flat, but rougher and in cross-section shows a wave-like appearance. On this side of the ribbons the grains are smaller and polygonal with larger precipitates, seen as dark dots. The wheel side of the ribbons has a faster cooling rate, which means the solidified melt is supersaturated. The TM alloying elements like Fe and Mn remain in a solid solution and the precipitates are much smaller. The grains are columnar and the growth direction is defined by the direction of the heat gradient, which is perpendicular to the copper wheel. The very fast cooling rate causes no precipitation during the crystal growth and this is the reason why the grains are larger. While the air-side ribbons solidify more slowly and the precipitates have more time to grow. This part of the ribbons is at a certain point still liquid and has many precipitates, which are seeds for the crystallization. That causes the formation of many small grains that begin to grow at almost the same time.

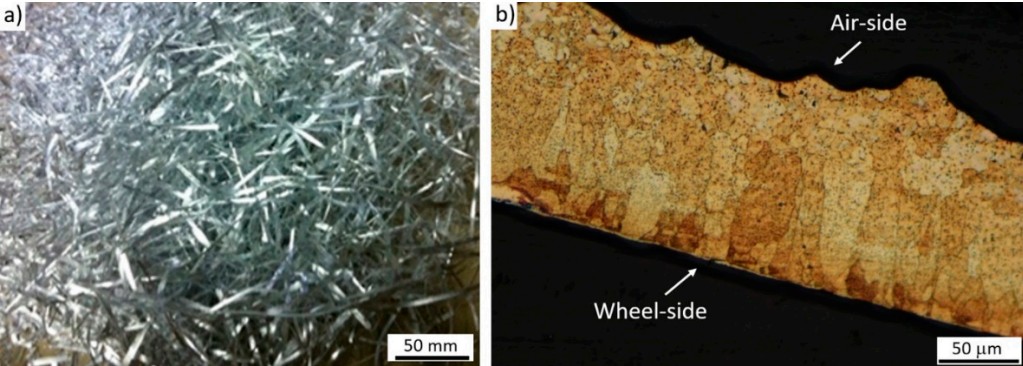

**Figure 2.** (**a**) Melt-spun ribbons, (**b**) longitudinal cross-section of ribbon (light microscopy (LM), etched by two-stage Weck).

SEM images reveal the microstructure of ribbons even better. Depending on the chemical composition, some ribbons have less or no precipitates, as shown in Figure 3a, where there is a cross-section of the ribbon with no addition of Fe and/or Mn. The addition of TM elements, Mn in Figure 3b and Mn and Fe in Figure 3c, causes more precipitation. As already described for Figure 2b, the majority of ribbons have larger precipitates on the

air side; however, it sometimes happens that the ribbons touch each other or the time of the contact with copper wheel can be different. Due to very uneven conditions during melt spinning, the large precipitates are not always on the air side.

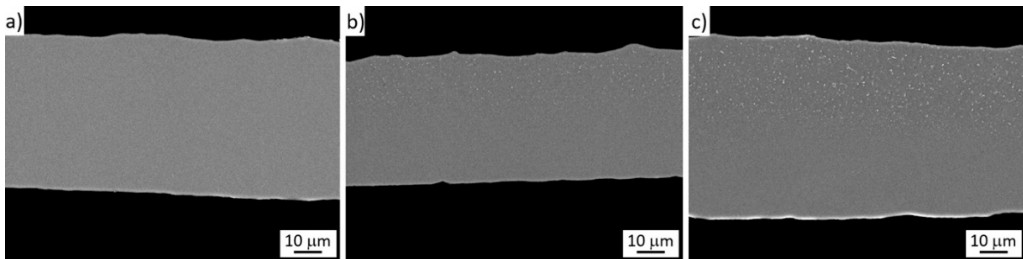

**Figure 3.** SE images of the cross-section of the melt-spun ribbons (**a**) rapid solidification (RS) AA 5083 with Mn, (**c**) RS AA5083 with Mn and Fe.

Figure 4 shows the air-side ribbon with bright and dark precipitates analyzed by SEM/EDS. In SE image, the areas of the EDS analyses are marked and the related chemical composition are presented in the inset table. Even though EDS analysis are rather inaccurate for small phases we assume that fine bright phases, according to our EDS analysis and the literature [3,15], are some of these possible phases; $\alpha$-Al$_{15}$(FeMn)$_3$Si, $\beta$-Al$_5$FeSi, Al$_9$Mn$_3$Si and/or $\alpha$-Al$_{12}$Fe$_3$Si. There are many of these bright colored phases in the microstructure. There is less dark phase (Mg$_2$Si), which is present only in the RS ribbons that were the most slowly cooled. All the phase sizes are submicron, mostly smaller than 500 nm. There are no large phases in the RS ribbons. In the inset table of EDS results in Figure 4 highlighted elements shows the possible bright (Spectra 1 and 2) and dark (Spectra 3 and 4) phases.

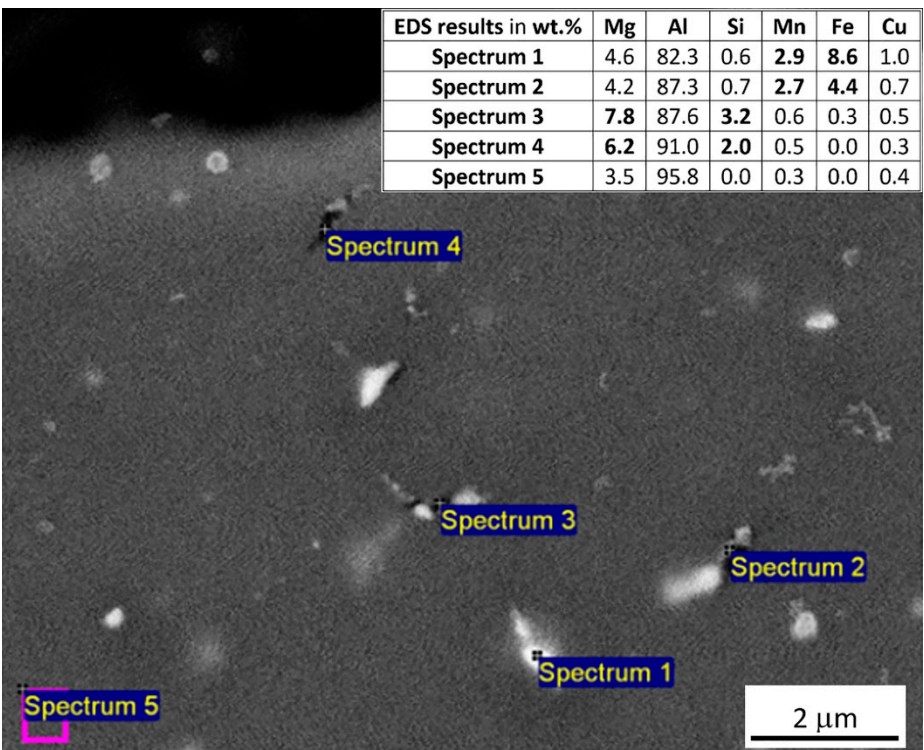

| EDS results in **wt.%** | Mg | Al | Si | Mn | Fe | Cu |
|---|---|---|---|---|---|---|
| Spectrum 1 | 4.6 | 82.3 | 0.6 | **2.9** | **8.6** | 1.0 |
| Spectrum 2 | 4.2 | 87.3 | 0.7 | **2.7** | **4.4** | 0.7 |
| Spectrum 3 | **7.8** | 87.6 | **3.2** | 0.6 | 0.3 | 0.5 |
| Spectrum 4 | **6.2** | 91.0 | **2.0** | 0.5 | 0.0 | 0.3 |
| Spectrum 5 | 3.5 | 95.8 | 0.0 | 0.3 | 0.0 | 0.4 |

**Figure 4.** SE micrograph with marked EDS analyses and corresponding EDS results.

TEM BF micrographs were recorded for all three studied samples and both wheel-side and air-side areas. In all the studied areas small precipitates were present and the amount of those precipitates increased with the addition of TM elements. TEM/EDS analyses were performed and in Figure 5 is a TEM BF micrograph with a lot of nanosized precipitates

of the sample RS AA5083 with Mn and Fe. In the TEM BF micrographs, large amounts of precipitates based on AlMnFe ($Al_6$(Mn,Fe) phase) are present in a spherical shape of 50 nm and needles of 50 nm thick and approximately 100–500 nm long. Those precipitates are formed because of the supersaturated TM elements in the rapidly solidified aluminum matrix. They were also studied by Stan-Glowinska et al. [3].

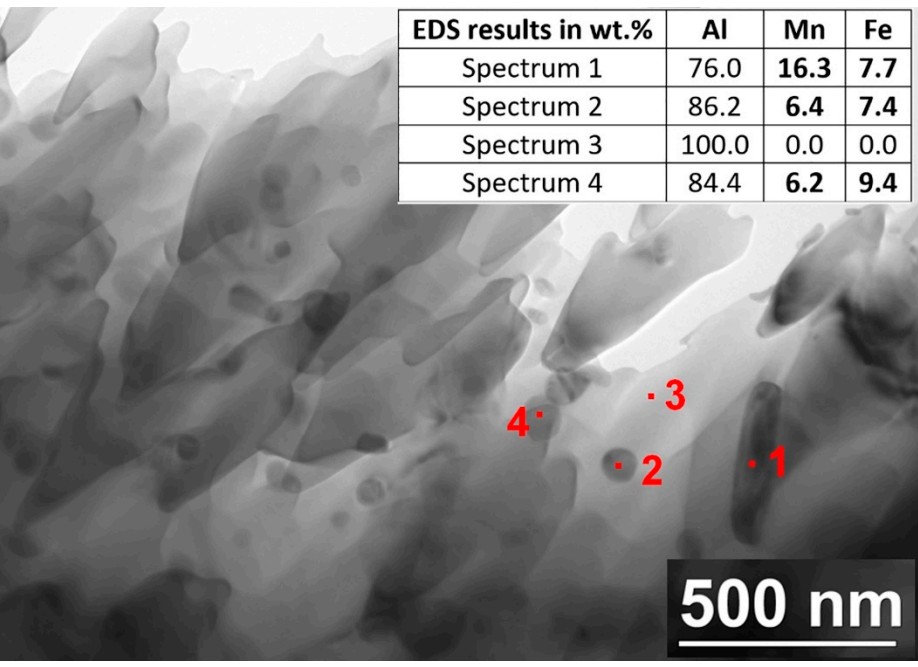

| EDS results in wt.% | Al | Mn | Fe |
|---|---|---|---|
| Spectrum 1 | 76.0 | **16.3** | **7.7** |
| Spectrum 2 | 86.2 | **6.4** | **7.4** |
| Spectrum 3 | 100.0 | 0.0 | 0.0 |
| Spectrum 4 | 84.4 | **6.2** | **9.4** |

**Figure 5.** TEM bright field (BF) micrograph with marked analyzed EDS areas and corresponding EDS results.

The grain morphologies of the as-melt-spun and the annealed ribbons are clearly seen from the EBSD inverse pole figure in the Z direction (IPF-Z) map (Figure 6), similar to the findings of Tewari [16] using different alloy. The as-melt-spun ribbons in contact with the rotating wheel have larger and columnar grains. The other part solidified subsequently and has a much finer, polygonal grain structure (Figure 6a). Because the solidification starts at a large number of seeds at the same time, a very fine grain structure appears, with polygonal shapes. Due to the different solidification conditions of the ribbons, the bimodal structure is obtained. In the literature [2], quite often the formation of fine grains is correlated with the wheel side, because this part of the ribbons is much more quickly cooled and solidified, which seems to be contradictory. The very flat surface of the ribbon is undoubtedly proof of wheel side, which is correlated with columnar grains.

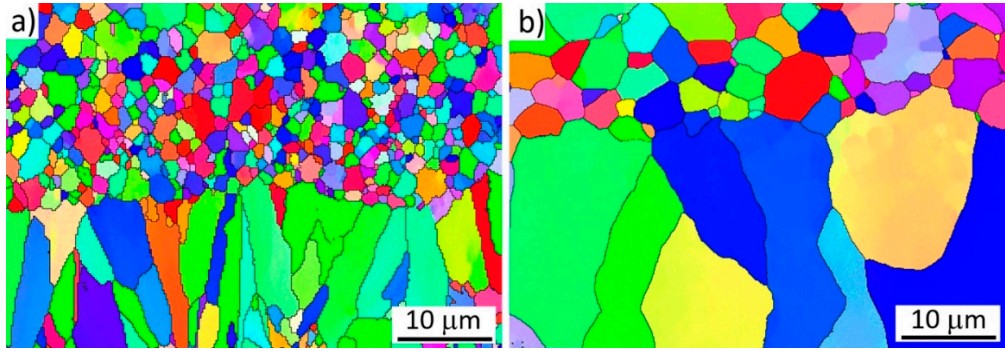

**Figure 6.** SEM electron back-scatter diffraction (EBSD) IPF-Z maps of cross-section of melt spun ribbons with addition of Mn and Fe, (**a**) as melt spun, (**b**) after annealing at 500 °C for 3 h.

The temperature stability of the ribbons was tested by exposure to different temperatures (T interval from 350 to 500 °C) and times (t = 3, 12, 24 h) and was performed to select the proper conditions for extrusion. No significant differences by measuring the microhardness were revealed up to 450 °C. The first drop in the microhardness of the ribbons was detected at 470 °C. The microstructure (Figure 6b) of the annealed ribbons at 500 °C for 3 h with the addition of Mn and Fe shows a slight difference in the grain shape, because they started to coarsen due to the high temperature and longer exposure time.

A comparison of the three RS ribbons with modified chemical compositions shows that the most temperature-resistant material is the one with the addition of Mn and Fe. Microhardness measurements of those ribbons show the highest values for all the temperature-testing ranges (Table 2).

**Table 2.** Vickers microhardness of ribbons (HV0.025) for each annealing temperature with three different annealing times.

| Annealing T (°C) | 350 | 400 | 450 | 470 | 500 | as-spun |
|---|---|---|---|---|---|---|
| Annealing t (h) | 3/12/24 | 3/12/24 | 3/12/24 | 3/12/24 | 3/12/24 | 3/12/24 |
| RS AA5083 | 96/87/87 | 96/83/82 | 89/84/81 | 82/81/81 | 79/81/75 | 98 |
| RS AA5083 + Mn | 97/97/96 | 98/95/96 | 95/94/92 | 87/83/84 | 82/79/78 | 105 |
| RS AA5083 + Mn + Fe | 107/104/104 | 101/102/101 | 98/96/95 | 89/89/87 | 87/85/83 | 109 |

A bimodal microstructure is achieved by rapid solidification due to the nature of the melt spinning's solidification and consists of two different regions: The wheel side and the air side. The wheel side is cooled faster and is composed of a supersaturated solid solution of TM elements and has larger columnar grains. On the other hand, the air side is less intensively cooled and consists of stable and metastable phases and has smaller grains, mainly under 1 μm. Voderova et al. [2,10] discuss the phases and precipitations achieved by melt spinning of aluminum alloys with addition of different TM elements. The rapidly solidified aluminum alloy with the addition of Fe (up to 11 wt. %) is composed of a supersaturated solid solution of Fe in aluminum and metastable $Al_6Fe$ [17] on the faster-cooling side, while the air side contains a smaller amount of stable $Al_{13}Fe_4$, also referred as $Al_3Fe$ [18], and metastable $Al_6Fe$. The supersaturated TM elements in our study were confirmed by TEM analyses in Figure 5, where all the nano-precipitates consist of Al, Mn and Fe. On the other hand, the air side of the ribbons contained $Al_{13}(Mn,Fe)_4$, $Al_{15}(Mn,Fe)_3Si$ and $Mg_2Si$ phases, which were analyzed by SEM/EDS in Figure 4.

After the melt-spun material was cold compacted into billets the hot extrusion was performed at 420 °C. Cross-sections perpendicular to the extrusion direction were prepared for EBSD microstructure analyses. Figure 7 shows the microstructure development during the extrusion of the RS ribbons. In Figure 7a is extruded RS AA5083 material with a homogeneous grain size and distribution. The grains of RS AA5083 with Mn (Figure 7b) and RS AA5083 with Mn and Fe (Figure 7c) show a bimodal gran structure where both larger and smaller grains in ribbons, despite hot extrusion, retain their size and shape. The effect of the bimodal structure appears due to the very good thermal stability of the RS material with the addition of TM elements. Nanoprecipitates based on TM elements keep the material stable by preventing the grain refinement of larger grains due to the mechanical deformation and grain growth due to the recrystallization temperature. The TM elements prevent the migration of dislocations and make it difficult for the sliding planes to move.

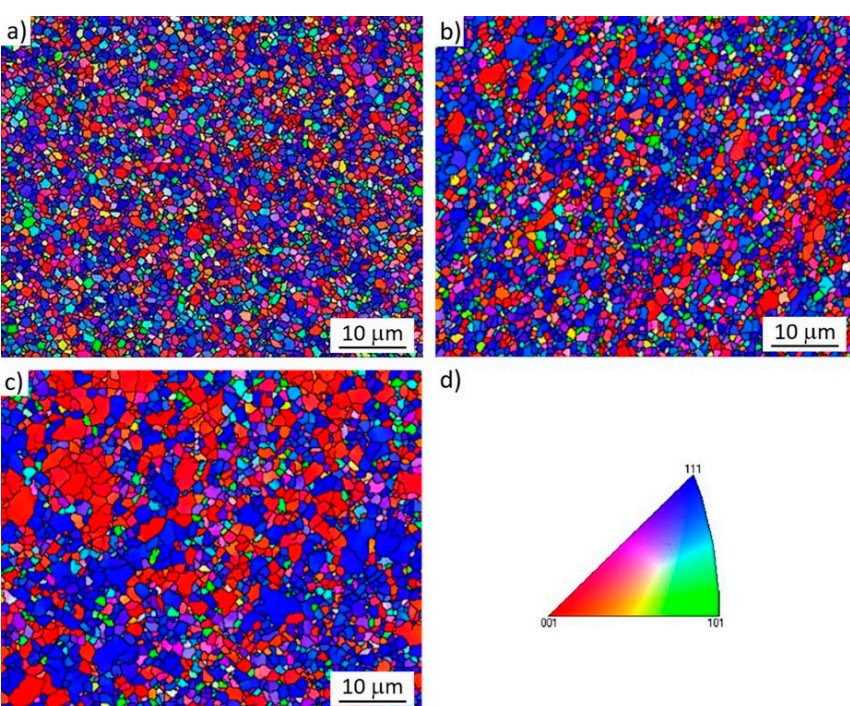

**Figure 7.** EBSD IPF Z maps of cross-section of the extruded material (**a**) IPF-Z of RS AA5083, (**b**) RS AA5083 with Mn, (**c**) IPF-Z of RS AA5083 with Mn and Fe, and (**d**) IPF legend.

The mechanical tests of the RS extruded material showed a significant improvement in the tensile and yield strength without decreasing the elongation. In comparison to the master alloy (industrially extruded) AA5083 material and the standard tensile strength of the extruded RS AA5083 increase for 5% and 30%, respectively, and for extruded RS AA5083 material with Mn and with Mn and Fe the increases were 20% and 48%, respectively, for Mn addition and 34% and 65% for Mn and Fe addition. Also, the yield strength of the extruded RS AA5083 with Mn and extruded RS AA5083 with Mn and Fe increased by 30% and 125% for the Mn addition and 45% and 151% for Mn and Fe additions, respectively. The elongation of the extruded RS AA5083 with Mn and Fe increased by 100% compared to the standard material and 14% compared to industrial extruded AA5083 master alloy. The average results of the mechanical tests are presented in Table 3.

**Table 3.** Mechanical properties of standard AA5083 compared with the studied materials.

| Samples | Tensile Strength $R_m$ (MPa) | Elongation A (%) | Yield Strength $R_{p0.2}$ (MPa) | HV0.5 |
|---|---|---|---|---|
| **Standard for AA5083 [14]** | 270 | 12 | 125 | 75 |
| **Extruded AA5083** | 334 | 20 | 216 | 90 |
| **Ex. RS AA5083** | 353 | 21 | 195 | 97 |
| **Ex. RS AA5083 with Mn** | 402 | 20 | 282 | 110 |
| **Ex. RS AA5083 with Mn and Fe** | 447 | 24 | 314 | 112 |

The potentiodynamic measurements of the four different extruded Al samples in a typical corrosion medium of 3.5% NaCl is shown in Figure 8 and Table 4. The corrosion rate calculations were performed according to the Faraday's law described in ASTM G102-89 (re-approved in 2015 standard) [19]:

$$v_{corr} = K_1 \, (i_{corr}/\rho) \, EW \tag{1}$$

where $K_1$ is 3.27 $10^{-3}$ mm/g μA cm yr (for mmpy units of $v_{corr}$), $i_{corr}$ is corrosion current density in μA/cm$^2$, which is calculated from the Stern–Geary equation [19] by $B/R_p$, where $B$ is calculated and $R_p$ is the polarization resistance from the slope of the potential versus current density plot taken approximately 20 mV on either side of the open circuit potential, ρ is density of the material in g/cm$^3$ and EW is equivalent weight (considered dimensionless in this equation), while $i_{corr}$ is in ohm-cm$^2$. Rp is the slope of the potential versus current density plot taken approximately 20 mV on either side of the open circuit potential where the slope is often approximately linear.

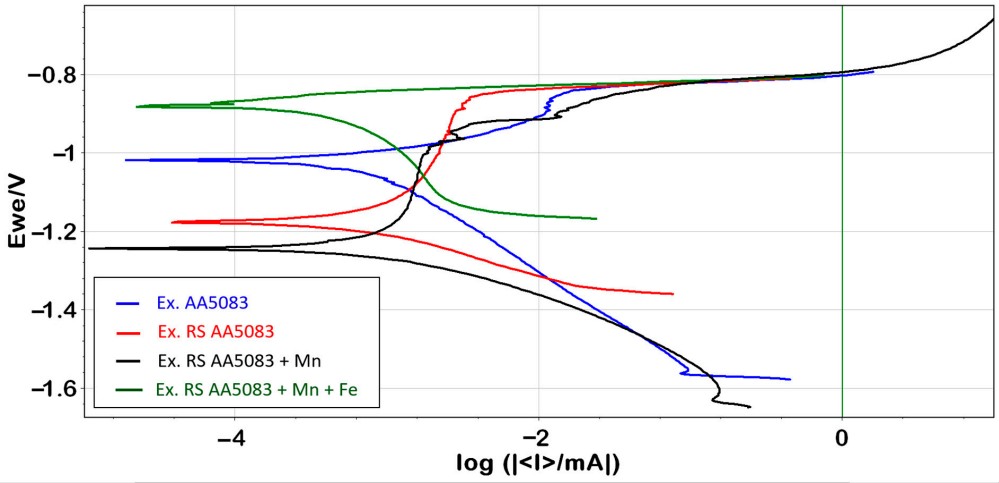

**Figure 8.** Polarization curves recorded for four different extruded RS Al samples in 3.5% NaCl.

**Table 4.** Corrosion properties of the studied extruded materials.

| Samples | $E_{corr\ vs.SCE}$ (V) | $i_{corr}$ (μA) | $v_{corr}$ (mm/y) | Corrosion Rate (mm/y) |
|---|---|---|---|---|
| Ex. AA5083 | −1.019 | 0.98 | $3.78 \times 10^{-2}$ | 0.0378 |
| Ex. RS AA5083 | −1.176 | 1.11 | $4.28 \times 10^{-2}$ | 0.0428 |
| Ex. RS AA5083 with Mn | −1.251 | 0.64 | $2.48 \times 10^{-2}$ | 0.0248 |
| Ex. RS AA5083 with Mn and Fe | −0.857 | 0.16 | $6.21 \times 10^{-3}$ | 0.0062 |

The main difference between the samples is the chemical composition and the rate of solidification (normal material extrusion and extrusion of RS material) of the alloys, which all affected the polarization potential and the passivation behavior. Before all the measurements of the corrosion parameters, 1 h of stabilization at the open-circuit potential (OCP) occurred. Corrosion potentials ($E_{corr}$) for these samples changed quite fairly, as the RS AA5083 with Mn has $E_{corr}$ around −1.2 $V_{SCE}$ and RS AA5083 with Mn and Fe sample has approximately −0.86 $V_{SCE}$. On the other hand, following the Tafel region, the alloys exhibited a differently broad range of passivation with practically the same breakdown potential ($E_b$) at around −0.83 $V_{SCE}$. The passivation range is significantly narrowed, and there is almost no passive region for sample RS AA5083 with Mn and Fe, whereas the corrosion-current density ($i_{corr}$) and corrosion rate ($v_{corr}$) was almost seven-times lower compared to the highest $i_{corr}$ and $v_{corr}$ of the sample RS AA5083. In the passive range the corrosion-current densities changed for the tested specimens from 0.16 mA to almost 1.11 mA. In the active-passive transition just two of the studied specimens went into the typical passive range, RS AA5083 and RS AA5083 with Mn, whereas the other two did not exhibit a typical passive region or it was considerably narrower.

The correlation between grain size and corrosion in the Al alloys has been extensively studied [20–23], but the literature data does not present a unified explanation for the effect of grain refinement on the corrosion resistance of Al alloys. On the other hand, the amount

of intermetallic inclusions usually plays a crucial role in the susceptibility of different Al alloys to corrosion characteristics [24,25]. However, the effects of grain boundaries and intermetallic phases on the corrosion are still not thoroughly understood and each play an important role in the final corrosion characteristics of the alloy.

The corrosion-favorable parameters in our studied Al alloys correlate with the grain size and consequently with the number/length of the grain boundaries where the inclusions less favorable to corrosion can form (e.g., β-phase ($Al_3Mg_2$)) with an increased amount of Mg that migrates to the grain boundaries [25]. In our case, the corrosion rate increases with the number of grains. As a result, our studied alloys corroborate with the data described for the similarly processed Al alloys (severe plastic deformation SVD, high-pressure torsion HPT, equal-channel angular pressing ECAP, friction steer welding FSW, etc.) [22,26]. The improved corrosion resistance in our study occurs in the samples with a bimodal microstructure. The EBSD data (Figure 7) show that the number of grains per analyzed area in the samples with a bimodal microstructure is reduced by approximately 50%. The smaller number of grains in the extruded RS AA5083 samples with the addition of Mn and Fe is due to partly the larger grains in the material's microstructure. The material with addition of Mn and Fe has larger amount of bimodal microstructure (Figure 7c) and also larger amount of grains with (001) planes parallel to the surface. From the literature [27] it is known that (001) single crystals has most noble pitting potential values. This can be explanation why material with Mn and Fe has despite the additional elements even better corrosion resistance.

## 4. Conclusions

The study was focused on the preparation of a high-strength aluminum alloy with superior mechanical properties and very good corrosion resistance at elevated temperature within the limits of the standard AA5083 alloy. By using the melt-spinning technique we achieved supersaturated, temperature-resistant ribbons from which the extruded aluminum material was produced. With the addition of transition-metal elements to the standard alloy we improved the mechanical and corrosion properties with a bimodal microstructure of the material.

The main achievements and conclusions are:

- The explanation of the grain morphology was related to the nature of the melt-spinning process—the grains are larger and columnar in the direction of the temperature gradient on the wheel side of the RS ribbons. On the other hand, the ribbons' air side consists of a larger amount of smaller grains, typically around 1 μm.
- The explanation of the precipitates was related to the nature of melt-spinning process—the precipitates on the wheel side of the RS ribbons are nano-sized, based on AlMnFe, due to the extremely high cooling rate. The air side of the RS ribbons consists of, besides the nano-sized AlMnFe precipitates, also micron-sized stable and metastable phases, like $Mg_2Si$, and phases based on AlMnFeSi.
- The relation of the chemical composition to the grain morphology after extrusion—the addition of transition-metal elements generates in the matrix material nano-sized precipitates, which influenced the grain behavior during extrusion. The material with more precipitates retains the bimodal structure due to melt spinning, independent of the subsequent deformation process, even at elevated temperatures. This makes the material stable at higher temperature.
- The relationship between the chemical composition and the mechanical and corrosion properties depends on the addition of transition-metal elements directly and indirectly influences the nano- and microstructure of the RS material, as well as the grain behavior during the thermal treatment. The optimal concentrations of TM elements improve all the mechanical properties and maintain or even improve the corrosion resistant of the RS aluminum alloy.

Based on experiments, we have shown that it is possible to produce aluminum alloys with high-strength properties and excellent corrosion resistance at elevated temperatures, which can also be introduced and up-scaled into an industrial manufacturing process.

**Author Contributions:** Conceptualization, I.P. and M.G.; methodology, I.P., M.G., and P.C.; software, I.P., and Č.D.; validation, I.P., M.G., and Č.D.; formal analysis, I.P., Č.D., and M.G.; investigation, I.P., M.G., and P.C.; writing—original draft preparation, I.P., M.G., and Č.D.; writing—review and editing, I.P. and M.G.; supervision, M.G. and I.P.; funding acquisition, P.C. and M.G. All authors have read and agreed to the published version of the manuscript.

**Funding:** This research was funded by the Slovenian Research Agency, research core funding No. P2-0132. This work was also funded by the project program ''Optimization of a new generation of high-strength corrosion-resistant aluminum alloys at elevated temperatures based on rapid solidification technology'', supported by IMPOL 2000 d.d. aluminum industry.

**Data Availability Statement:** The data presented in this study are available on request from the corresponding author.

**Conflicts of Interest:** The authors declare no conflict of interest.

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
