# Peer review of "Bimodal Microstructure Obtained by Rapid Solidification to Improve the Mechanical and Corrosion Properties of Aluminum Alloys at Elevated Temperature"

_metals, doi:10.3390/met11020230_

Round 1
Reviewer 1 Report
This manuscript includes and reports manufacturing process of aluminum alloy with modification of alloying element. The author analyzed the microstructure of aluminum alloy with various methods, including SEM, EBSD ... In addition, appropriate discussion was included in proper place. During my revision, I couldn't feel any uncomfortable with errors in manuscript. So, I strongly recommend to publish this manuscript in materials.
Author Response
Dear Editor and reviewer
Thank you very much for your time and the careful review of our manuscript. We very much appreciate your opinion of the study and the manuscript and we hope it will be published as you recommended.
Faithfully yours,
Irena Paulin
Comments:
This manuscript includes and reports manufacturing process of aluminum alloy with modification of alloying element. The author analyzed the microstructure of aluminum alloy with various methods, including SEM, EBSD ... In addition, appropriate discussion was included in proper place. During my revision, I couldn't feel any uncomfortable with errors in manuscript. So, I strongly recommend to publish this manuscript in materials.

Reviewer 2 Report
In the manuscript the authors studied the microstructure, mechanical properties, and corrosion resistance of aluminium alloysbAA5083. To obtain the good mechanical properties, this alloy was rapidly solidified using the met-spinning technique and then the ribbons were plastically consolidated by extrusion at elevated temperature.
The manuscript is interesting but must be significantly improved.
Comments:
- Detailed analysis of the SEM EBSD measurements should be performed. It is necessary to compare the grain orientation spread (GOS) values for melt spun, after annealing and for extruded material (Fig.6, 7). This parameter has influence on the corrosion resistance of some alloys. In Fig. 7 (c), the grains which have the crystallographic orientation close to 001 are not uniformly distributed. It should be explained.
- Different solidification conditions of ribbons cause the formation of bimodal microstructure. How this microstructure influences the corrosion behaviour of AA5083 alloy. If the corrosion tests have been performed on the wheel side and air side.
- The SEM and TEM analysis have revealed the presence of nanoprecipitates especially for the specimens where the TM have been added (Mn, Fe). How is the influence of this precipitates on the corrosion susceptibility of AA5083 alloy. Some polarization curves LSV revealed that the specimens are susceptible to pitting corrosion. Where the corrosion starts? Where are the week points?
- The graph which shows the evolution of OCP vs. time should be presented for all specimens.
- In the table 4 the authors have shown the corrosion parameters obtained for the extruded materials. It is necessary to explain how these parameters were obtained and calculated. The extrapolation of the Tafel line is not correct in the case of materials which undergo the passivation. Please check the limitation of the use the Tafel equation in the literature (Landolt, D., Corrosion and Surface Chemistry of Metals, EPFL Press, 2007). The presented analysis is not correct. For some LSV curves there is no Tafel slope (passive range). Therefore it is not possible to perform the extrapolation of the cathodic and anodic branch. The equation for calculation of corrosion rate should be given.
- The improved corrosion resistance occurs for specimen with bimodal microstructure. This must be explained why?
- The week point of the manuscript is lack of discussion how is the correlation between the microstructure and corrosion resistance of aluminium alloy.
Author Response
Dear editor and reviewer:
Thank you very much for your careful review and constructive suggestions with regard to our manuscript “Bimodal microstructure obtained by rapid solidification to improve the mechanical and corrosion properties of aluminium alloys at elevated temperature”. We have studied comments carefully and tried our best to revise and improve the manuscript according to the reviewers’ valuable comments. The main corrections in the paper are marked yellow and the responds to the reviewer’s comments are as following:
Comments:
- Detailed analysis of the SEM EBSD measurements should be performed. It is necessary to compare the grain orientation spread (GOS) values for melt spun, after annealing and for extruded material (Fig.6, 7). This parameter has influence on the corrosion resistance of some alloys. In Fig. 7 (c), the grains which have the crystallographic orientation close to 001 are not uniformly distributed. It should be explained.
R1: Thank you very much for your comment regarding the EBSD measurements. In this research we used EBSD to analyse the grain size, because the grains are so small and are not so well analysed using light microscopy of the etched samples. This was the reason for our decision to preform EBSD analysis, where small grains are nicely seen. The orientation and distribution of the individual grains were not part of this study. We are also aware that orientation can influence the corrosion behaviour of materials (we added some explanation at the end of the section Results and Discussion), but in our study we wanted to confirm that the corrosion resistance (of the otherwise corrosion-resistant Al alloy) is not decreasing, although the other properties (e.g., mechanical) are being improved.
Why the (001) grains are not uniformly distributed is very difficult to explain at this stage. We are planning to perform an experiment where during the extrusion process the tool with material will be quickly cooled and we will study the microstructure during the extrusion process, which will give us an answer about the microstructure and the texture development.
- Different solidification conditions of ribbons cause the formation of bimodal microstructure. How this microstructure influences the corrosion behaviour of AA5083 alloy. If the corrosion tests have been performed on the wheel side and air side.
R2: The corrosion tests were performed on four different extruded samples and the intention of those tests was to show that although the chemical composition of AA5083 is being modified and the rapid solidification followed by plastic deformation (extrusion) was use for the material production, the corrosion resistance is not being aggravated, it has even been improved. The corrosion tests were not performed on individual ribbons on the wheel or air side, because that was not a goal of the study. Technically, it is not possible to perform the corrosion tests on a particular side of the ribbon (wheel or air) with the same testing conditions as were used for the extruded materials. The corrosion was tested only on the manufactured materials.
- The SEM and TEM analysis have revealed the presence of nanoprecipitates especially for the specimens where the TM have been added (Mn, Fe). How is the influence of this precipitates on the corrosion susceptibility of AA5083 alloy. Some polarization curves LSV revealed that the specimens are susceptible to pitting corrosion. Where the corrosion starts? Where are the week points?
R3: Thank you for raising this question. Indeed, the nanoparticles can influence the corrosion resistance (we mentioned the influences on corrosion on page 8) and our main attention was to perform corrosion tests on the manufactured material and to see if the corrosion rate is increasing. However, from the results of the corrosion tests (that were performed under the same conditions for all the tested specimens) we show that the corrosion resistance is not aggravated, but rather improved. The goal of the research was to improve the mechanical properties of the standard aluminium alloy and improve the resistance at elevated temperatures. Improving the corrosion resistance can be interesting for an additional extended study.
- The graph which shows the evolution of OCP vs. time should be presented for all specimens.
R4: Thank you for your comment. In this study we extensively investigated the mechanical properties and properties at elevated temperatures. The corrosion was studied only to show that despite the chemical composition changes (adding TMs), the corrosion parameters remain similar or are even improved. We described that the OCP have been performed 1 h for the stabilisation before the measurements and it has been used just for the stabilisation of the samples and unfortunately did not perform the OCP vs. time measurements. This will be performed in our next extended study of corrosion properties in aluminium alloys. Thank you very much for you valuable suggestions.
- In the table 4 the authors have shown the corrosion parameters obtained for the extruded materials. It is necessary to explain how these parameters were obtained and calculated. The extrapolation of the Tafel line is not correct in the case of materials which undergo the passivation. Please check the limitation of the use the Tafel equation in the literature (Landolt, D., Corrosion and Surface Chemistry of Metals, EPFL Press, 2007). The presented analysis is not correct. For some LSV curves there is no Tafel slope (passive range). Therefore it is not possible to perform the extrapolation of the cathodic and anodic branch. The equation for calculation of corrosion rate should be given.
R5: Thank you for this observation that we have mistakenly omitted the equation for the calculated corrosion rate. We corrected the manuscript accordingly to the ASTM G102 standard for corrosion calculations. Regarding the extrapolation of the Tafel lines we agree with your comment that in some cases in our study Tafel extrapolation is on the limit where the extrapolation is still reasonable, but we still think that for the reason of the present study, where we compare the produced material, this extrapolation is sufficient. Thank you again for this valuable comment and we will in the next extended corrosion study perform also other corrosion measurements, like linear polarisation, cyclic voltammetry, EIS.
- The improved corrosion resistance occurs for specimen with bimodal microstructure. This must be explained why?
R6: Thank you for your comment. On page 8 we explained why the bimodal structure is better for corrosion resistance (fewer grain boundaries due to slightly larger grains): ‘’The improved corrosion resistance in our study occurs in the samples with a bimodal microstructure. The EBSD data (Figure 7) show that the number of grains per analysed area in the samples with a bimodal microstructure is reduced by approximately 50 %. The smaller number of grains in the extruded RS AA5083 samples with the addition of Mn and Fe is due to partly the larger grains in the material’s microstructure.’’
- The week point of the manuscript is lack of discussion how is the correlation between the microstructure and corrosion resistance of aluminium alloy.
R7: As we already mentioned, the main goal of the study was not the corrosion behaviour of the material but an improvement of the mechanical properties at elevated temperature while keeping good corrosion properties (as the aluminium alloys of 5xxx series already have good corrosion resistance). Nevertheless, we interpreted the results correlated between the obtained microstructure of the extruded samples and its corrosion resistance with a discussion that is also confirmed in the literature (page 8, the last two paragraphs before Conclusions).
Regarding your comments about lack of discussion about the correlation between microstructure and the corrosion resistance of the aluminium alloy, we added at the end of the Results and Discussion section, where we are explaining the corrosion in terms of the grain orientations.
Once again, we appreciate for reviewers’ time and warm work earnestly, and hope that the corrections will meet with approval.
Yours sincerely,
Irena Paulin
Reviewer 3 Report
Very nice manuscript which seems to be an original. The text has a good structure, and it is clear to read. It is dealing with the microstructural, mechanical and corrosion properties of AA5083 alloys which were prepared by rapid solidification followed by extrusion at 420 °C. Adding of transition-metal elements Mn and Fe up to 1 and 0.45 wt. %, respectively, seems to be beneficial for improvement of corrosion resistance, tensile strength, hardness and even elongation of aluminium alloys.
The results are sufficiently discussed in the text. I have only several minor comments and questions which should be revised before publication.
Page 3, Table 1: For faster orientation of reader in the article, it would be better to mention in description that also chemical composition of standard alloy is in the table included.
Page 4, Line 167-142: Use subscript in the abbreviations: Icorr, Ecorr, Rm, Rp0.2
Page 5, Line 184: “…there are many of them”. How can you be so sure? The EDS analysis of so small particles is rather inaccurate. You should also mention in text what does the bold highlighted chemical composition in the table of the Figure 4 mean.
Page 7, Table 2: Three annealing times for as-spun alloys listed in the table are confusing.
According the template of the journal, the reference numbers should be placed in square brackets [ ] in the text. Please, use also the prescribed formatting of references (name of journals with normal letters).
Author Response
Dear editor and reviewer:
Thank you very much for your careful review and constructive suggestions with regard to our manuscript “Bimodal microstructure obtained by rapid solidification to improve the mechanical and corrosion properties of aluminium alloys at elevated temperature”. We have studied the comments carefully and tried our best to revise and improve the manuscript according to the valuable comments. The main corrections in the paper are marked in yellow and the responses to the reviewer’s comments are written below.
Comments:
Very nice manuscript which seems to be an original. The text has a good structure, and it is clear to read. It is dealing with the microstructural, mechanical and corrosion properties of AA5083 alloys which were prepared by rapid solidification followed by extrusion at 420 °C. Adding of transition-metal elements Mn and Fe up to 1 and 0.45 wt. %, respectively, seems to be beneficial for improvement of corrosion resistance, tensile strength, hardness and even elongation of aluminium alloys.
The results are sufficiently discussed in the text. I have only several minor comments and questions which should be revised before publication.
- Page 3, Table 1: For faster orientation of reader in the article, it would be better to mention in description that also chemical composition of standard alloy is in the table included.
R1: Thank you for your comment. We checked the manuscript and it is written on page 2, the first sentence in the last paragraph on that page: ‘’The chemical compositions of the standard AA5083 aluminium alloy and the modified samples with the addition of only Mn as well as Fe and Mn are presented in Table 1.’’
- Page 4, Line 167-142: Use subscript in the abbreviations: Icorr, Ecorr, Rm, Rp0.2
R2: Thank you for noticing and reminding us to unify the subscripts. They have been corrected in the text (yellow marked).
- Page 5, Line 184: “…there are many of them”. How can you be so sure? The EDS analysis of so small particles is rather inaccurate. You should also mention in text what does the bold highlighted chemical composition in the table of the Figure 4 mean.
R3: Thank you for your very valuable comment. We performed analyses on many areas of the samples and it is clear from the SE images that there are many fine light (bright) phases in the matrix. We are not confirming the exact stoichiometry of the phases, but only from literature known possible phases according to the approximation of the EDS results. We added to the manuscript (yellow marked part on page 5) the highlighted chemical composition, which we found helpful to see the increased amount of chemical elements at the analysed points.
- Page 7, Table 2: Three annealing times for as-spun alloys listed in the table are confusing.
R4: Thank you for the comment. The table caption of Table 2 has been supplemented with an explanation of the presented results.
- According the template of the journal, the reference numbers should be placed in square brackets [ ] in the text. Please, use also the prescribed formatting of references (name of journals with normal letters).
R5: Thank you for your comment. We corrected the references accordingly.
Once again, we appreciate for reviewers’ warm work earnestly, and hope that the corrections will meet with approval.
Yours sincerely,
Irena Paulin
Round 2
Reviewer 2 Report
The authots should only add the explanation how the icorr was dtermined.
Author Response
Thank you for you comment about icorr determination. We added the explanation how the icorr was determined in the manuscript (page 8, marked in yellow).
Kind regards,
Irena Paulin